# Inappropriate Use of Emergency Services from the Perspective of Primary Care Underutilization in a Local Romanian Context: A Cross-Sectional Study

**DOI:** 10.3390/healthcare12070794

**Published:** 2024-04-06

**Authors:** Anca Maria Lăcătuș, Ioana Anisa Atudorei, Andrea Elena Neculau, Laura Mihaela Isop, Cristina Agnes Vecerdi, Liliana Rogozea, Mihai Văcaru

**Affiliations:** 1Department of Fundamental, Clinical and Prophylactic Sciences, Transylvania University, 500036 Brașov, Romania; anca.lacatus@unitbv.ro (A.M.L.); laura.isop@unitbv.ro (L.M.I.); r_liliana@unitbv.ro (L.R.); mihai.vacaru@unitbv.ro (M.V.); 2Department of Social Sciences and Communication, Transylvania University, 500036 Brașov, Romania; ioana.atudorei@unitbv.ro; 3Department of Clinical and Surgical Disciplines, Transylvania University, 500036 Brașov, Romania; cris_vecerdi@unitbv.ro

**Keywords:** primary care, underutilization, emergency services, out-of-hours primary care

## Abstract

***Background***: The underutilization of primary care services is a possible factor influencing inappropriate emergency service presentations. The objective of this study was to evaluate the proportion and characteristics of patients inappropriately accessing emergency room services from the perspective of primary care underutilization. ***Methods***: This cross-sectional study included patients who visited the emergency room of a County Hospital, initially triaged with green, blue, or white codes, during a 2-week period in May 2017. Two primary care physicians performed a structured analysis to correlate the initial diagnosis in the emergency room with the final diagnosis to establish whether the patient’s medical complaints could have been resolved in primary care. ***Results***: A total of 1269 adult patients were included in this study. In total, the medical problems of 71.7% of patients could have been resolved by a primary care physician using clinical skills, extended resources, or other ambulatory care and out-of-hours services. ***Conclusions***: Low awareness of out-of-hours centers and a lack of resources for delivering more complex services in primary care can lead to inappropriate presentations to the emergency services. Future research on this topic needs to be conducted at the national level.

## 1. Introduction

Various factors underlying patients’ use of emergency room (ER) services for non-urgent conditions have been identified in the literature [1]. Some of these factors are associated with patients’ perceptions of their medical situation, their relationship with primary care physicians (PCPs), and their ease of access to primary care services [2]. The use of ERs for non-urgent services is driven by patients’ misconceptions that their conditions are serious and require urgent care when they do not [3,4,5]. Patients also perceive ERs as convenient access points for medical care, particularly specialist care and examinations such as laboratory and imaging tests, which can be performed on the same day and in the same location [3,5,6,7,8]. A lack of access to primary healthcare, either caused by the lack of a regular primary healthcare provider in the area, the inaccessibility of the provider owing to restricted visiting hours or logistical reasons, such as transport and childcare, also leads to improper use of ER services [5,9,10,11,12]. The inaccessibility of a prescribing physician and the need to renew prescriptions have also been identified as the most frequent reasons for non-urgent visits [13]. A lack of knowledge regarding primary healthcare and out-of-hours services and the available treatment options delivered by these providers are further drivers of ER misuse. Additionally, a lack of health literacy, specifically a failure to understand the purpose of ERs, what constitutes an emergency, and a lack of general knowledge regarding health and illness, exists [14,15]. Another factor underpinning the overuse of ERs is patients’ perception that they will receive a better standard of care in ERs than from their primary healthcare provider due to their dissatisfaction with the level of care received from their primary healthcare provider [16,17].

Primary healthcare services are generally self-employed, mainly solo practices, with an average patient list of 1800 and an average working time of 35 h/week, paid through a mixture of capitation and fee-for-service payments. There is a deficit of doctors, especially in rural areas. The “Health at a Glance” European report, published in 2016, showed that an average of 27% of patients across the European Union (EU) visited an ER because primary care was not available. This proportion is paradoxically low in Romania, where <15% of patients reported accessing ERs because of the lack of availability of their usual primary care providers [18].

Primary care continues to be underutilized; the phenomenon of patients seeking medical care by reporting directly to hospitals for non-urgent conditions is presented in the Country Health profile of Romania in the 2023 State of Health in the EU report, demonstrating that no major changes in the organization of healthcare have been made [19].

Although peer-reviewed local data showing the burden of non-urgent presentations in ERs could not be identified, a 2018 report from the Department for Emergency Situations in Romania reported that ERs were accessed 5.38 million times in the previous year [20]. Of these, only <17.8% of cases had red and yellow triage codes (indicating a high level of severity and the need for immediate intervention), whereas the rest had green, blue, and white triage codes that required lower-complexity interventions. Observational studies have indicated that 10–49.6% of patients presenting to emergency departments can be managed with primary care [2,21].

An alternative point of patient access to primary care in Romania is out-of-hours primary care centers (OOH-PCs). OOH-PCs are financed by the Ministry of Health and are practise-based centers, self-driven by groups of five to seven doctors, with direct access (no triage) irrespective of insurance status and possess opening hours outside the working times of PCPs, including 24/24 at weekends. Although they have been established in Romania since 2004 [22], most patients do not seem to be aware of OOH-PC services (Box 1). According to a report from the National Health Insurance House (NHIH) in 2020 [23], medical care in OOH-PCs was solicited by 1,159,429 patients, representing a 36% increase from 2019. However, these services remain underutilized, and reports on their actual utilization for the years 2021–2022 are not available from the NHIH.

This study aimed to evaluate the proportion and characteristics of patients accessing ER services with conditions that could have been resolved in primary care.

## 2. Methods

### 2.1. Study Design, Settings and Outcomes

This was a cross-sectional monocentric study. It was performed in a county emergency tertiary hospital serving a medium-sized county (comprising several cities as well as rural areas) with a population of around 630,000 people. 

Main outcomes 

The main outcome was the post hoc possibility to resolve the case outside the ER. The 5 possible categorical values are “yes, by PCP with clinical skills alone”, “yes, by PCP with clinical skill and point-of-care resources”, “yes by PCP with supplementary outpatient services”, “no”, and cannot be categorized. 

The other outcome that was investigated was patients’ awareness of the existence of OOH-PCs. Possible responses were “yes, I was aware” and “no, I was not aware”. 

### 2.2. Definitions

The Ministry of Health’s Ordinance nr 443/2019 that concerns the national protocol of triage in emergency departments [24] codes an incoming ER patient in one of 5 categories—red, yellow, green blue, and white—as presented in Table 1. 

### 2.3. Selection of Participant

Inclusion criteria: Adult patients (over 18 years of age) triaged with green, blue, and white codes who consented to participate and completed an episode of care in the ER.

Exclusion criteria: Patients triaged with red and yellow codes, involved in road accidents and aggressions, those with alcohol abuse issues, patients addressed for non-medical issues (social cases), and patients refusing participation.

### 2.4. Data Collection Method

A standardized paper-based checklist for data collection was employed. This checklist comprised two sections. The triage nurse filled out the first section at the initial triage area. Separately, two experienced PCP study investigators (AML, AEN) independently completed the second section after the episode of care concluded.

The triage nurses included only patients who consented to the study and were classified in green, blue, or white categories during the initial standard triage in the ER, following the national protocol.

The first section included data regarding demographics (age, gender, educational level, residential environment, medical insurance status, and registration with a PCP), prior contact with a PCP for the ongoing episode, means of reaching the ER, time elapsed from symptom onset to ER presentation, awareness of OOH-PCs existence, clinical information like the disease code according to the International Classification of Primary Care, 2nd edition (ICPC-2), required resources to conclude the case (laboratory tests or interdisciplinary consults), and a subjective evaluation by us, the investigators, of the case’s manageability in an OOH-CC.

In the second section, the post hoc solvability of cases outside the ER was assessed, based on the investigators’ comprehensive understanding of PCPs in Romania. This took into consideration the curricular training of a PCP and the best practice recommendations for patient care available in Romania. Patients were categorized based on the required medical care into three groups: those resolvable with a PCP’s clinical examination, those needing clinical examination and extensive resources, and those requiring clinical examination plus outpatient investigations or consultations. The “PCP with extensive resources” category encompasses PCPs with access to a variety of point-of-care tests. Atypical cases, such as mismatches between the presentation and the diagnostic result together with those culminating in hospitalization were considered as unsolvable by PCPs, regardless of the initial triage code.

### 2.5. Data Analysis

Categorical data are presented as frequencies and percentages, and continuous variables as means with standard deviations (SD). The Chi-square test was employed to examine differences between groups of categorical values and ANOVA to examine differences between categories of continuous values. Spearman’s r was used to determine the strength of correlations. The interobserver reliability for the main outcome was assessed using Cohen’s *kappa* index, which measures the degree of agreement between raters beyond chance. Missing data were identified, and our analyses were limited to cases with complete information.

An adjusted standardized residual above the 1.96 threshold for a 95% confidence interval indicates a significant difference from expected frequencies under the null hypothesis of no association [25].

All statistical analyses were conducted using IBM SPSS Statistics for Windows, Version 26.0, Armonk, NY, USA: IBM Corp. Results with a *p*-value of less than 0.05 were considered as statistically significant.

### 2.6. Ethical Approval

The study protocol followed the principles of the Declaration of Helsinki, and the research checklist was approved by the Ethics Committee of the County Hospital (registration number: No. 14) on 8 May 2017. The participants were provided with information about the study’s aims and procedures, the confidentiality and anonymity of the collected data, the completely voluntary nature of their participation, and whether their participation or non-participation would affect the standard of medical care they would be receiving. Informed consent was obtained from all participants.

## 3. Results

In this study, 1269 patients fulfilled the inclusion criteria and were subsequently analyzed (refer to Table 2).

Among the non-critical cases, upon their arrival at the ER, most were assigned green codes, accounting for 81.88% of the cases. A majority of these patients were from urban settings (67.3%) and had attained a medium level of education (34.83%). Regarding the timing of ER visits, a notable pattern emerged: a majority (38.46%) sought ER services significantly after the onset of their symptoms, indicating a delay in seeking emergency care (as detailed in Table 3).

The medical issues of a majority of patients, 71.6% (n = 910), could have been resolved by a PCP (Table 2). The distribution of the possibility of a PCP managing the case in pre-hospital conditions across different triage codes is summarized in Table 2. 

Chronic diseases, new-onset diseases, and minor accidents were the main reasons for ER presentations, with some distinct possibilities for each category to be managed outside the ER (*p* < 0.001 **).

A higher proportion (60.9%) of younger patients’ (18–39 years) medical problems could have been solved by a PCP in their offices, 27.6% only by clinical examination and 33.3% requiring point of care tests, compared to the cases of patients aged over 60 years (36.4%) (Table 3). In this study, it was observed that an increase in the age of the applicants corresponded to an elevated requirement for care from a PCP with access to extensive resources.

It was also noted that 82.19% of participants had not contacted their PCP before coming to the ER, and only 4.96% were aware of out-of-hours primary care services (*p* = 0.049 **). The level of education or residence in urban or rural areas did not impact their awareness. 

The association between the type of disease (chronic/acute/accidents) patients were diagnosed with upon ER admission and their potential for receiving care either outside the ER or solely within it was also investigated. The analysis revealed a significant association based on the condition type (χ^2^ = 46.09, *p* < 0.001), indicating that the likelihood of a patient being treatable by a PCP or requiring ER attention varies with their condition. In a subgroup analysis, the results showed that patients with new-onset disease could be treated by a PCP with or without access to extended resources, in their offices; patients presenting with accidents and trauma could be treated by PCP with ambulatory services or ER, and patients with chronic diseases likely needed ER treatment.

The association between patients’ awareness of OOH-PC and the possibility of being treated outside the ER is confirmed by a Chi-square test (χ^2^ = 7.85, *p* = 0.04). There is an associational relationship between people who said they had never heard of OOH-PC and the fact that their condition could be treated by a family doctor with extensive resources (adjusted standardized residual = 2.8).

The day of the week also impacted the patient flow, with easier-to-manage patients presenting in the middle of the week (adjusted standardized residual = 4 for an association of cases manageable with only clinical means on Thursday) (Table 4).

Agreement on Case Management Outside the ER: A calculated Cohen’s Kappa index of 0.81 demonstrated a high degree of agreement between investigators on the potential for case management outside the ER. 

## 4. Discussion

The first hypothesis in this study was: “A majority of ER presentations coded as green, blue or white could be resolved in primary care”. The rate of inappropriate presentations to the ER in our study was 71.6%, supporting the first hypothesis. This is higher than the range reported in other studies [4,5,26,27,28,29]. However, this difference could be a result of the various methods used to assess urgent and non-urgent cases.

The results highlight specific problems within the national healthcare system, such as the underfinancing of primary care. According to the Country Health Profile of Romania in the 2021 State of Health in the EU report, 44% of the health budget in Romania was allocated to inpatient care in 2019, which was the highest proportion in the EU, where the average is 29%. This situation is continuing to date, as stated in the 2023 State of Health in the EU report, despite the political programs in place to strengthen primary care [19,30]. Only 18.6% of health expenditure was allocated to primary and ambulatory care, which is the second-lowest proportion in the EU. This comparative overallocation of resources to inpatient care resulted in underspending in other sectors of healthcare, such as primary or preventative care.

The choice to access ER services without first engaging with the primary care system may be underpinned by a lower level of access to primary care compared with other countries in the EU. According to the Country Health Profile of Romania in the 2021 State of Health in the EU report, local PCPs have a lower average working time of 35 h per week compared with other European countries, where the average is 45 h per week. The availability of medical doctors in Romania is also lower than that in the EU, with 3.2 practicing doctors per 1000 people compared with the EU average of 3.9 practicing doctors per 1000 people. This trend was also corroborated specifically for PCPs: 24.5% of the doctors’ workforce comprised PCPs in 2019, which is below the EU average of 26.5%. Addressing access to primary care services may reduce the burden on ER departments. For example, Bruni et al. (2016) concluded that increasing primary care accessibility for up to 12 h/day could reduce inappropriate ER presentations by 10–15% [31]. Similarly, Dolton and Pathania (2016) reported that the availability of PCP services 7 days a week reduced weekly ER presentations by 9.9%, with a 17.9% reduction during weekends [32].

The second hypothesis in this study was: “A majority of patients are unaware of OOH-PC services”. The results indicate that 89.76% of participants were unaware of OOH-PC services, despite PCPs working in a continuity-of-care system in OOH-PCs being able to offer extended access to care. These findings support the second hypothesis. 

The third hypothesis in this study was: “There is a statistical relationship between inappropriate ER presentations and lack of awareness of OOH-PC”. A significant association was identified between the possibility of resolving the case outside the ER and patients’ awareness of OOH-PCs. A positive correlation between the possibility of resolving the case outside the ER and awareness of OOH-PCs indicated that a lack of awareness of the availability of OOH-PCs can be a significant factor influencing inappropriate presentation to the ER. This finding underscores the necessity for raising awareness and developing educational programs that can guide potential patients to appropriate healthcare services.

A secondary factor in patients’ choices of medical services may be the complexity of their clinical problems. This choice depends on the patient’s perception of the severity and urgency of their condition [1,33]; however, it can also be influenced by the availability of equipment for diagnosis and treatment in the primary care setting. In the present study, 27.9% of non-urgent cases could have been treated by PCPs with extended resources, such as additional skills, electrocardiography, and point-of-care tests. This finding suggests that the potential solutions to the issue of inappropriate ER presentations could be the development of support skills for PCPs and equipping PCP offices with more extensive resources.

A significant correlation was also identified between patient age and the possibility of their clinical case being treated outside the ER. Younger patients were more likely to be treatable by a PCP alone, whereas older patients were more likely to require extensive clinical care. This may result either from a lack of awareness regarding the availability of OOH-PC, or simply be due to the convenience of accessing clinical evaluation and treatment in the ER (no appointment necessary, centralized investigation and treatment, wide range of investigations available, etc.) Therefore, the younger population represents a more relevant target for informative and educational campaigns regarding the appropriate use of ER services. However, further in-depth studies are warranted to explore the reasons underlying young individuals’ inappropriate use of ER services.

## 5. Limitations

The data collection, limited to a single County Hospital over two weeks, restricts the generalizability of our findings. Nonetheless, this study offers valuable insights for future research on this topic, both within Romania and internationally. Given the data were gathered in 2017 and the absence of significant changes in the regulatory or organizational framework of primary care since then, it is unlikely that patient behavior towards ER use has significantly shifted.

## 6. Conclusions

A considerable majority of non-urgent patients in this study could have received care in a primary care setting. Various factors, including the availability and capacity of PCPs to conduct comprehensive patient evaluations, influence the choice to bypass primary care in favor of direct ER access. The interrater reliability of two independent evaluators showed a high concordance, suggesting that the criteria used to assess whether a case could be managed by a PCP are reliable and could be standardized for broader applications. The study underscores the importance of educating the population on the judicious use of health services, the appropriate contexts for ER visits, and, crucially, the availability of OOH-PC services. However, to gain a clearer understanding of the situation and identify specific areas for improvement in primary care, national-level longitudinal studies are essential.

## Figures and Tables

**Table 1 healthcare-12-00794-t001:** The Romanian National Triage protocol in the emergency room.

Red code (resuscitation): Patients who require life-saving intervention. Maximum access delay to the treatment area: 0 min. Yellow code (critical): Patients who present with a high-risk situation, altered mental status (acute change), or any intense pain or major discomfort. Maximum access delay to the treatment area: 10 min. Green code (urgent): Patients with stable vital functions but who require two or more medical resources. Maximum access delay to the treatment area: 30 min. Blue code (non-urgent): Patients with stable vital functions who require a single medical resource. Maximum access delay to the treatment area: 60 min. White code (consultation): Patients who do not require emergency medical care or any medical resources and present at the ER for vaccination, as a social case without clinical complaints, and for clinical-administrative problems (medical certificates, prescriptions). Maximum access delay to the treatment area: 120 min.

**Table 2 healthcare-12-00794-t002:** Sociodemographic characteristics of the participants and their patterns of presentation to the emergency room. n = 1269.

Characteristic	Frequency	Could Be Solved by PCP Clinically	Could Be Solved by PCP with POC Resources	Could Be Solved by PCP with Aditional External Resources	Could Not Be Solved by PCP	*p*
	N (% of total)	n of category (% of category)	n of category (% of category)	n of category (% of category)	n of category (% of category)	
Age category						<0.001 **
18–39 years	425 (33.49)	117 (27.6)	141 (33.3)	83 (19.6)	83 (19.6)	
<40–59 years	384 (30.26)	96 (25.1)	116 (30.3)	79 (20.6)	92 (24)	
Over 60 years	460 (36.25)	71 (15.5)	96 (20.9)	111 (24.2)	181 (39.4)	
Mean age (years) (+/− SD)	50.12 (18.79)	45.48 (16.95)	45.93 (17.47)	52.00 (18.43)	56.70 (19.66)	<0.001 *
Gender						0.035 **
Female	584 (46.02)	139 (23.8)	180 (30.8)	120 (20.5)	145 (24.8)	
Male	672 (52.96)	143 (21.3)	170 (25.4)	151 (22.5)	206 (30.7)	
Residential environment						0.155
Urban	857 (67.53)	180 (21)	250 (29.2)	182 (21.3)	244 (28.5)	
Rural	351 (27.66)	88 (25.1)	85 (24.3)	84 (24)	93 (26.6)	
Education						0.545
Primary education	143 (11.27)	33 (23.1)	43 (30.1)	32 (22.4)	35 (24.5)	
Professional education	192 (15.13)	40 (20.8)	48 (25)	52 (27.1)	52 (27.1)	
High school	250 (19.70)	63 (25.3)	80 (32.1)	49 (19.7)	57 (22.9)	
University degree	175 (13.79)	40 (23)	59 (33.9)	38 (21.8)	37 (21.3)	
NR	509 (40.11)					

NR, no report; PCP, primary care physician; OOH-PC, out-of-hours primary care center. * ANOVA; ** Chi square.

**Table 3 healthcare-12-00794-t003:** Clinical characteristics of the patients (n = 1269).

Characteristic	Frequency	Could Be Solved by PCP Clinically	Could Be Solved by PCP with POC Resources	Could Be Solved by PCP with Aditional External Resources	Could Not Be Solved by PCP	*p*
	N (% of total)	n of category (% of category)	n of category (% of category)	n of category (% of category)	n category (% of categoy)	
Distribution of patients according to the triage code						<0.001 **
Green	1039 (81.88)	201 (19.4)	287 (27.6)	229 (30.9)	321 (30.9)	
Blue	197 (15.52)	74 (37.6)	64 (32.5)	34 (17.3)	25 (12.7)	
White	6 (0.47)	3 (50)	0	1 (16.7)	2 (33.3)	
Not coded	27 (2.10)					
Characteristics of presentation in the ER						<0.001 **
Chronic aggravated disease	344 (27.11)	72 (20.9)	78 (22.7)	83 (24.1)	111 (32.3)	
New-onset disease	642 (50.59)	178 (27.8)	198 (30.9)	116 (18.1)	149 (23.2)	
Minor accidents and trauma	258 (20.33)	30 (11.7)	70 (27.2)	70 (27.2)	87 (33.9)	
NR	25 (1.97)					
Contacted a PCP before presentation						<0.001 **
No	1043 (82.19)	246 (23.6)	303 (29.1)	218 (20.9)	274 (26.3)	
Yes	133 (10.48)	21 (15.8)	24 (18.0)	40 (30.1)	48 (36.1)	
Not registered with a PCP	26 (2.05)					
NR	67 (5.28)					
Knowledge about OOH-PCs						0.049 **
Yes	63 (4.96)	16 (25.4)	8 (12.7)	17 (27.0)	22 (34.9)	
No	1139 (89.76)	257 (22.6)	327 (28.8)	243 (21.4)	310 (27.3)	
NR	67 (5.28%)					
Brought by an ambulance						<0.001 **
Yes	343 (27.03)	56 (16.3)	74 (21.6)	68 (19.8)	145 (42.3)	
No	602 (47.44)	140 (23.3)	187 (31.1)	158 (26.2)	117 (19.4)	
NR	324 (25.53)					
Admitted to the hospital after being brought by an ambulance (n = 343)						<0.001 **
Yes	75 (21.87)	12 (7.2)	13 (7.8)	52 (31.3)	89 (53.6)	
No	253 (73.76)	263 (25)	322 (30.6)	214 (20.3)	255 (24.2)	
NR	15(4.37%)					
Time elapsed between symptom onset and ER presentation						<0.001 **
3 h	377 (29.71)	59 (15.7)	114 (30.4)	56 (14.9)	146 (38.9)	
3–24 h	389 (30.65)	95 (24.4)	131 (33.7)	66 (17.0)	97 (24.9)	
Over 24 h	488 (38.46)	130 (26.6)	105 (21.5)	148 (30.3)	105 (21.5)	
NR	15 (1.18)					

NR, no report; ER, emergency room; PCP, primary care physician; OOH-PC, out-of-hours primary care center. ** Chi square.

**Table 4 healthcare-12-00794-t004:** Daily distribution of cases.

Characteristic	Frequency	Could Be Solved by PCP Clinically	Could Be Solved by PCP with POC Resources	Could Be Solved by PCP with Aditional External Resources	Could Not Be Solved by PCP	*p*
		n of category (% of category)	n of category (% of category)	n of category (% of category)	n of category (% of category)	
Day of the week	n = 1266	284	353	273	356	<0.001 **
Monday	157	38 (13.4)	51(14.4)	34 (12.5)	34 (9.6)	
Tuesday	157	30 (10.6)	34 (9.6)	39 (14.3)	54 (15.2)	
Wednesdey	182	38 (13.4)	50 (14.2)	42 (15.4)	52 (14.6)	
Thursday	201	67 (23.6)	41 (11.6)	26 (9.5)	67 (18.8)	
Friday	177	31 (10.9)	45 (12.7)	54 (19.8)	47 (13.2)	
Saturday	221	36 (12.7)	77 (21.8)	40 (14.7)	68 (19.1)	
Sunday	171	44 (15.5)	55 (15.6)	38 (13.9)	34 (9.6)	

NR, no report; PCP, primary care physician; OOH-PC, out-of-hours primary care center. ** Chi square.

## Data Availability

Research data are available at the following link: https://osf.io/ufhxq/?view_only=2cda714dd1ae433f9d7bd4d191053b0f (accessed on 25 January 2024).

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
