# Peer review of "Inappropriate Use of Emergency Services from the Perspective of Primary Care Underutilization in a Local Romanian Context: A Cross-Sectional Study"

_healthcare, 2024, doi:10.3390/healthcare12070794_

Round 1

Reviewer 1 Report

Comments and Suggestions for Authors

Thank you for submitting the manuscript for review.

The study is very interesting and evaluates the use of emergency departments in a regional hospital in Romania. Over 71% of the cases in the emergency rooms could have been treated by a family doctor/primary care specialist. The authors discuss, among other things, the poor reachability and limited resources in primary care (e. g.; lack of rapid tests?).

I have a few comments that should be taken into account in the revision.

Introduction

1)     The introduction is very well presented.

2)     Please, make a hypothesis. What did you hypothesize?

Methods

3)     What does "a regional hospital" mean?  What does the population of 634,236 mean? Is this a large city or a metropolis in Romania?

4)     Data Collection Methods: AML and AEN are authors? Please add the information.

5)     “…Patients who consented…”? Or consented to the triage? Surely, you mean consent to the study? Could you please rephrase?

6)     Please put a space between [24] and (Table 1)

7)     Table 1: Please adjust “The National Triage Protocol of Romania”. Because in other countries, blue means "without survival, dying".

8)     Please ensure that the formatting of the font and tables is consistent, e.g. sufficient space between Table 1 and the text.

9)     Lines “For a more rigorous analyses…”.

10) The meaning of clinical resources is not entirely clear to me. Are clinical resources understood as diagnostics in the hospital (possibly hospitalization?)? Or is it the clinical resources at the family doctor? Please formulate the classification a little more clearly. What do you mean by clinical resources?

Results

11) Was the day of the week of presentation at the hospital taken into account? Can general practitioners be closed at the weekend?  Would that also be a reason to go to the hospital?

12) Line “….ER and the time elapsed until arrival at the ER…” From what? The travel time to the emergency room or from the onset of symptoms to the emergency room?

Discuss

13) Discuss medical care in the rural environment. Obviously there are big differences here.

14) How is the health competence of the population assessed, especially that of younger people? Is it possible that health competence decreases with younger age? Can this explain the higher frequency?

Limitation

15) See comment no. 11. Would that be a limitation?

Conclusions

16) What suggestions for improvement do you have? What should be done to reduce the number of cases in emergency departments?

Reviewer 2 Report

Comments and Suggestions for Authors

The study presented by Lacatus et all, describes the inappropriate use, in a rural area of Romania, of the emergency room.

The study is interesting but there are some important lacks.

Major

Methods

In data analysis please review the entire paragraph, move the software information to the end of the paragraph and create a paragraph apart with all the variables in the study. 

Quantitative variables must be summarized with the opportune index of descriptive statistics and they should be reported also in Table 2.

Since you have a lot of variables also in Table 3 please add the categories for each qualitative variable (for example "possibility of resolving the case outside ER" yes or no) 

The word correlation has a precise value in Statistics with chi-square you explore if there are associations, not correlations.

Please add a reference or explanation of Goodman and Kruskal's gamma test, it will be better if you report a description and the type of variables used for each test.

Results

Table 3 is completely unclear, it would be better to build a new one.

What do you mean with the type of test "Adjusted residual value"? You never report details, please clarify and add in case.

On page 6 please make more clear the entire paragraph under the table.

It will be also more scientifically sound if you create a profile about the patients with inappropriate use of ER and the others to have more details about their socio-demographic and clinical characteristics.

Minor

In Table 2 please add the sample size

In Table 3 what do you mean with NR? Please add the extension.

Round 2

Reviewer 1 Report

Comments and Suggestions for Authors

Thank you for revising the manuscript. Thank you for discussing my comments.  I have one more comment.

Please move the table heading of table 3 to the side of the table.

Reviewer 2 Report

Comments and Suggestions for Authors

Dear Authors,

Thanks for your reply, but I have some other comments that will be necessary to complete your work.

Since it is a scientific work it would be optimal to make everything impersonal and always refer to the study not to who conducted it (e.g. "The study showed an..." or "The result from this sample showed..." or "Data were summarized as mean and standard deviation for...".

In the aim "We hypothesized that most patients are unaware of OOH-PC services and that a majority of ER presentations coded as green, blue or white could be resolved in primary care without ER presentation. We also hypothesize that there is a statistical relationship between inappropriate ER presentations and lack of awareness of OOH-PC" This sentence is better to review and put in the discussion because the previous version of the aim was better.

From a clinical point of view one of these statements "(adjusted standardized residual. = 2.8)." has no "clinical" significance for readers. If you want to leave please detail better the potential impact of this information. Please make this for all the statements about adjusted std residual.

Please in data analysis move the last sentence before the software statements and add a reference to this "criteria" that you choose.

"This suggests that criteria used to assess whether a case could be managed by a PCP are reliable and could be standardized for broader applications." This is not a result but a comment please move to Discussion/Conclusion.

I don't find this information in Table 3 "A higher proportion of younger patients (18-39 years) medical problems could have been solved by a PCP in their office (60.9%), compared to the cases of patients aged over 60 years (36.4%) (Table 3). " I think you have to correct with Table 2.

Please review this sentence "We investigated the association between the type of disease (chronic/acute/accidents) patients were diagnosed with upon ER admission and their potential for receiving care either outside the ER or solely within it" since you are writing results (e.g. "The evaluation between... showed that...")

Please in Table 2 correct n=1269 and not 1296

Please report the value of p-value even if it was not significant.

All the "N" if they regarding a sample will be better to report as "n"
